# Long-term exposure to air pollution and severe COVID-19 in Catalonia: a population-based cohort study

Otavio Ranzani [1,2,3], Anna Alari[1,2,3], Sergio Olmos[1,2,3], Carles Milà[1,2,3], Alex Rico[1,2,3], Joan Ballester [1], Xavier Basagaña [1,2,3], Carlos Chaccour [1,4,5], Payam Dadvand [1,2,3], Talita Duarte-Salles [6], Maria Foraster[1,2,3,7], Mark Nieuwenhuijsen[1,2,3], Jordi Sunyer[1,2,3], Antònia Valentín [1,2,3], Manolis Kogevinas[1,2,3], Uxue Lazcano [8,9], Carla Avellaneda-Gómez[10], Rosa Vivanco[9] & Cathryn Tonne [1,2,3] ✉

The association between long-term exposure to ambient air pollutants and severe COVID-19 is uncertain. We followed 4,660,502 adults from the general population in 2020 in Catalonia, Spain. Cox proportional models were fit to evaluate the association between annual averages of $PM_{2.5}$, $NO_2$, BC, and $O_3$ at each participant's residential address and severe COVID-19. Higher exposure to $PM_{2.5}$, $NO_2$, and BC was associated with an increased risk of COVID-19 hospitalization, ICU admission, death, and hospital length of stay. An increase of $3.2\,\mu g/m^3$ of $PM_{2.5}$ was associated with a 19% (95% CI, 16–21) increase in hospitalizations. An increase of $16.1\,\mu g/m^3$ of $NO_2$ was associated with a 42% (95% CI, 30–55) increase in ICU admissions. An increase of $0.7\,\mu g/m^3$ of BC was associated with a 6% (95% CI, 0–13) increase in deaths. $O_3$ was positively associated with severe outcomes when adjusted by $NO_2$. Our study contributes robust evidence that long-term exposure to air pollutants is associated with severe COVID-19.

Ambient air pollution is a main contributor to the global burden of disease, including cardiovascular and respiratory diseases[1]. Although there is extensive literature on the effects of short- and long-term exposure to ambient air pollution on chronic respiratory diseases[2], evidence is limited for long-term exposure and incidence or severity of acute respiratory infections[3].

COVID-19, caused by infection by the SARS-CoV-2 virus, mainly presents as an acute respiratory infection. Several risk factors have been identified for progression to severe disease and mortality, such as age, male sex, and chronic comorbidities[4]. It is well known that air pollutants, both particulate matter and gases, can impair lung defenses against infections[5]. Additionally, there is evidence showing the potential effect of air pollutants upregulating the expression of SARS-CoV-2 receptors in the lung[6]. Early in the pandemic, ecological studies reported associations between ambient air pollution and increased risk of hospitalization and death by COVID-19[7]. However, individual-level cohort studies are needed to overcome the multiple methodological limitations of ecological studies on the topic[7,8]. A number of individual-level studies reported positive associations between long-term exposure to air pollutants and hospital admission or death,

[1]Barcelona Institute for Global Health, ISGlobal, Barcelona, Spain. [2]Universitat Pompeu Fabra (UPF), Barcelona, Spain. [3]CIBER Epidemiología y Salud Pública (CIBERESP), Madrid, Spain. [4]Universidad de Navarra, Pamplona, Spain. [5]CIBER Enfermedades Infecciosas (CIBERINFEC), Madrid, Spain. [6]Fundació Institut Universitari per a la recerca a l'Atenció Primària de Salut Jordi Gol i Gurina (IDIAPJGol), Barcelona, Spain. [7]PHAGEX Research Group, Blanquerna School of Health Science, Universitat Ramon Llull (URL), Barcelona, Spain. [8]Instituto Biodonostia, Grupo Atención Primaria, San Sebastian, Spain. [9]Agency for Health Quality and Assessment of Catalonia (AQuAS), Barcelona, Spain. [10]Complex Hospitalari Moisès Broggi, Consorci Sanitari Integral, Barcelona, Spain. ✉e-mail: cathryn.tonne@isglobal.org

particularly for fine particulate matter [PM with an aerodynamic diameter of ≤2.5 μm] ($PM_{2.5}$) but less consistently for nitrogen dioxide ($NO_2$). These studies followed cohorts of positive COVID-19 cases[9–11] or selected populations[12–15], and one analyzed the general population[16]. However, several knowledge gaps remain due to the heterogeneity in observed estimates for COVID-19 severity[12,17] and death[9,10,12,18], likely because of the limited sample size and the number of events in previous studies, and the lack of multi-pollutant models.

To address these evidence gaps, we analyzed a large population-based cohort of the general population in Catalonia. We investigated associations between $PM_{2.5}$, $NO_2$, ozone ($O_3$) and black carbon (BC) and hospital and intensive care unit (ICU) admission, hospital length of stay, and death related to COVID-19 during 2020.

## Results
### Population and exposure characteristics
Figure 1 shows the course of the COVID-19 pandemic during 2020 in Catalonia, Spain. The study flowchart is shown in Supplementary Fig. S1. From 4,669,011 adult individuals alive and residing in Catalonia on March 1, 2020, we excluded 409 (<0.1%) because of loss to follow-up, 589 (<0.1%) because of inconsistent dates, 1512 (<0.1%) missing residential address and 5999 (0.1%) missing air pollutants exposure values, resulting in 4,660,502 individuals included in our analyses.

In 2020, there were 340,608 COVID-19 diagnoses, of which 216,752 (64%) were laboratory confirmed. The majority of COVID-19 diagnoses occurred at the primary care units (249,878; 73%). Among the 340,608 cases, there were 47,174 (14%) COVID-19-related

hospitalizations, 4699 (1.4%) ICU admissions, and 10,001 COVID-19-related deaths (3%). Among the 10,001 deaths, 3744 (37%) occurred among non-hospitalized individuals. The median hospital LOS was 7 [p25–p75: 4–14] days. The description of the COVAIR-CAT cohort and COVID-19-related events is shown in Table 1.

Annual averages (SD) of air pollution in the cohort were 13.9 (2.2) μg/m³ for $PM_{2.5}$, 26.2 (10.3) μg/m³ for $NO_2$, and 91.6 (8.2) μg/m³ for $O_3$ from the COVAIR-CAT 2019 models. The distribution of these pollutants' concentrations and the exposure estimates from the ELAPSE models and their correlations are shown in Supplementary Methods.

### Associations with COVID-19 severe events
In single-pollutant models (Main Model–Model 4, Table 2 and Fig. 2), higher annual average exposure to $PM_{2.5}$ and $NO_2$ was associated with a greater hazard of COVID-19-related events. For $PM_{2.5}$, there were positive associations for hospitalization (HR 1.19, 95% CI, 1.16–1.21), ICU admission (HR 1.16, 95% CI, 1.09–1.24), and death (HR 1.13. 95% CI, 1.07–1.19) per IQR increase. For $NO_2$, there were positive associations for hospitalization (HR 1.25, 95% CI, 1.22–1.29), ICU admission (HR 1.42, 95% CI, 1.30–1.55), and death (HR 1.18, 95% CI, 1.10–1.27) per IQR increase. For both $PM_{2.5}$ and $NO_2$, positive associations were observed for hospital LOS. In two-pollutant models, $NO_2$ remained positively associated with hospital and ICU admission after adjustment for $PM_{2.5}$. Similarly, positive associations for $PM_{2.5}$ remained for hospital admission and hospital LOS after adjustment for $NO_2$. For $O_3$, the association was negative for COVID-19-related events in single-pollutant models and null or positive when co-adjusted for $NO_2$: HR

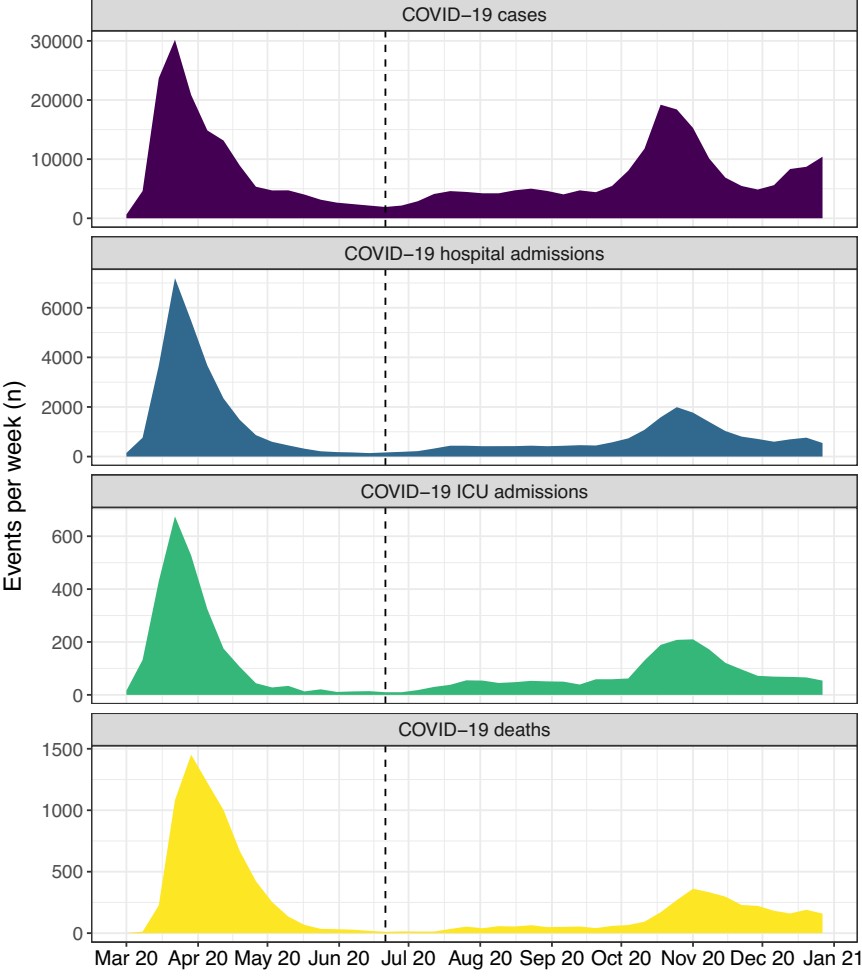

**Fig. 1 | Weekly COVID-19 cases and severe related events during 2020 in Catalonia, Spain.** The vertical dashed black line refers to the limit between the first and second waves (June 21, 2020).

**Table 1 | Characteristics of the cohort overall and according to COVID-19 outcomes**

| | Overall | COVID-19 hospital admission | COVID-19 ICU admission | COVID-19 death |
|---|---|---|---|---|
| n | 4,660,502 | 47,174 | 4699 | 10,001 |
| **Age**, years, mean (SD) | 53.6 (17) | 65.7 (17) | 63.3 (12) | 81.7 (10) |
| **Female**, n (%) | 2,446,855 (52.5) | 22,288 (47.2) | 1508 (32.1) | 5149 (51.5) |
| **Tobacco smoking**, n (%) | | | | |
| Non-smoker | 3,033,731 (65.1) | 31,911 (67.6) | 2878 (61.2) | 6943 (69.4) |
| Former smoker | 680,895 (14.6) | 10,057 (21.3) | 1242 (26.4) | 2254 (22.5) |
| Active smoker | 945,876 (20.3) | 5206 (11.0) | 579 (12.3) | 804 (8.0) |
| **Individual income group**, n (%) | | | | |
| Low | 3,240,314 (69.5) | 34,119 (72.3) | 3229 (68.7) | 7909 (79.1) |
| Middle | 1,393,153 (29.9) | 12,826 (27.2) | 1441 (30.7) | 2055 (20.5) |
| High | 27,035 (0.6) | 229 (0.5) | 29 (0.6) | 37 (0.4) |
| **Health risk group**, n (%) | | | | |
| Healthy | 2,334,035 (50.1) | 11,681 (24.8) | 1250 (26.6) | 567 (5.7) |
| Low | 1,394,963 (29.9) | 13,601 (28.8) | 1563 (33.3) | 1903 (19.0) |
| Moderate | 698,598 (15.0) | 13,133 (27.8) | 1252 (26.6) | 3838 (38.4) |
| High | 232,906 (5.0) | 8759 (18.6) | 634 (13.5) | 3693 (36.9) |
| **Chronic comorbidities**, n (%) | | | | |
| Diabetes mellitus | 471,419 (10.1) | 11,959 (25.4) | 1350 (28.7) | 3731 (37.3) |
| Obesity | 1,160,099 (24.9) | 19,701 (41.8) | 2341 (49.8) | 3927 (39.3) |
| COPD | 223,500 (4.8) | 6128 (13.0) | 557 (11.9) | 2116 (21.2) |
| Hypertension | 1,181,252 (25.3) | 22,578 (47.9) | 2229 (47.4) | 6839 (68.4) |
| Other cardiovascular disorders | 364,787 (7.8) | 9538 (20.2) | 785 (16.7) | 3662 (36.6) |
| Dyslipidemia | 1,305,896 (28.0) | 20,539 (43.5) | 2209 (47.0) | 5119 (51.2) |
| Nursing home status | - | 4433 (9.4) | 128 (2.7) | 2933 (29.3) |
| **Area of residence indicators** | | | | |
| **Urbanicity**, n (%) | | | | |
| City | 2,893,786 (62.1) | 33,434 (70.9) | 3287 (70.0) | 6630 (66.3) |
| Town and suburb | 1,360,492 (29.2) | 11,021 (23.4) | 1122 (23.9) | 2740 (27.4) |
| Rural | 406,224 (8.7) | 2719 (5.8) | 290 (6.2) | 631 (6.3) |
| **Socioeconomic indexes** | | | | |
| Small area socioeconomic index, median [IQR] | 41.05 [32.20, 49.45] | 41.71 [32.69, 50.90] | 42.32 [33.72, 51.26] | 40.78 [32.16, 49.50] |
| Deprivation index, z-score, median [IQR] | −0.54 [−1.04, −0.04] | −0.52 [−1.03, 0.02] | −0.45 [−0.98, 0.09] | −0.60 [−1.09, −0.06] |
| Percentage of non-Spanish residents, %, median [IQR] | 11.8 [7.0, 18.2] | 12.5 [7.6, 19.3] | 13.2 [8.0, 20.4] | 12.0 [7.3, 18.2] |
| Gini index, median [IQR] | 29.5 [27.2, 32.3] | 29.6 [27.2, 32.2] | 29.6 [27.3, 32.2] | 29.7 [27.3, 32.4] |
| **Health access** | | | | |
| Distance to closest primary care unit, meters, median [IQR] | 422 [262, 644] | 409 [258, 600] | 397 [252, 587] | 422 [264, 632] |
| Average weekly TPP, %, median [IQR] | 9.16 [7.95, 10.01] | 9.19 [8.09, 10.01] | 9.19 [8.09, 10.14] | 9.16 [8.09, 10.14] |

Higher values of the small area socioeconomic index denote a worse socioeconomic position compared to the average of Catalonia; higher values of the deprivation index denote more deprivation Compared with the average of Spain.
*COPD* chronic obstructive pulmonary disease, *TPP* test-positive proportion.

1.10 (95% CI, 1.02–1.18) for ICU admission and 1.01 (95% CI, 0.95–1.07) for death per IQR in $O_3$. Regarding hospital LOS, $O_3$ was positively associated with hospital LOS in two-pollutant models (Supplementary Table S1, Supplementary Fig. S2). Unadjusted estimates are shown in Supplementary Table S2. Results per one-unit increase in air pollution are shown in Supplementary Table S3.

All associations were comparable with Model 4 (main model) in sensitivity analyses, except when including cases diagnosed at nursing homes and evaluating COVID-19 deaths (Fig. 2; Supplementary Figs. S2, S3, S4, and S5; Supplementary Tables S4, S5, and S6). When evaluating associations by wave, the estimated measures of effect for the first wave were of greater magnitude than for the second wave for hospitalization (Table 3, Supplementary Table S7). The majority (80.4%) of hospital admissions had COVID-19 mentioned as a cause of hospital admission. The association of long-term exposure to air pollutants with hospitalization had also slightly greater magnitude for COVID-19-

related hospitalization defined by COVID-19 or respiratory causes, or COVID-19 only, as main causes of admission, compared to all-cause admissions (Table 4, Supplementary Tables S8 and S9).

When evaluating the subset with COVID-19 diagnosis, the results were consistent with the main analysis for hospitalizations and ICU admission, although of a smaller magnitude than the main analysis for $NO_2$ and $PM_{2.5}$. The associations with death were null in the whole period while positive in the second wave for $NO_2$ and $PM_{2.5}$ (Supplementary Tables S10, S1, and S12). Overall, there were no associations for $O_3$, except positive associations for death during the first wave. Effect estimates based on the COVAIR-CAT exposure models for 2018 and ELAPSE-2010 were broadly comparable to those in the main analyses (Supplementary Tables S13 and S14). For BC, there were positive associations for hospitalizations (HR 1.19, 95% CI, 1.16–1.22), ICU admissions (HR 1.19, 95% CI, 1.10–1.28), deaths (HR 1.06, 95% CI, 1.00–1.13) and hospital LOS (IRR 1.04, 95% CI, 1.02–1.07).

**Table 2 | Adjusted associations between long-term air pollutants and COVID-19-related outcomes in single and two-pollutant models**

| | Exposure | COVID-19 hospital admission*<br>HR (95% CI) | COVID-19 ICU admission*<br>HR (95% CI) | COVID-19 death*<br>HR (95% CI) | Hospital length of stay*<br>IRR (95% CI) |
|---|---|---|---|---|---|
| COVAIR-CAT models | | | | | |
| **NO$_2$** (IQR increase: 16.1) | Single-pollutant | 1.25 (1.22–1.29) | 1.42 (1.30–1.55) | 1.18 (1.10–1.27) | 1.06 (1.03–1.09) |
| **PM$_{2.5}$** (IQR increase: 3.2) | Single-pollutant | 1.19 (1.16–1.21) | 1.16 (1.09–1.24) | 1.13 (1.07–1.19) | 1.06 (1.04–1.08) |
| COVAIR-CAT models | | | | | |
| **NO$_2$** (IQR increase: 16.1) | Adjusted for PM$_{2.5}$ | 1.12 (1.08–1.17) | 1.51 (1.33–1.72) | 1.10 (0.99–1.22) | 0.99 (0.95–1.03) |
| **NO$_2$** (IQR increase: 16.1) | Adjust for O$_3$ | 1.24 (1.19–1.29) | 1.58 (1.39–1.79) | 1.19 (1.08–1.31) | 1.07 (1.03–1.12) |
| **PM$_{2.5}$** (IQR increase: 3.2) | Adjusted for NO$_2$ | 1.12 (1.08–1.15) | 0.93 (0.85–1.03) | 1.08 (1.00–1.16) | 1.07 (1.04–1.10) |
| **PM$_{2.5}$** (IQR increase: 3.2) | Adjust for O$_3$ | 1.16 (1.13–1.19) | 1.13 (1.04–1.22) | 1.12 (1.05–1.19) | 1.07 (1.05–1.10) |

*The analyses of COVID-19 hospital admission, ICU admission, and death were conducted in the whole population, while hospital length of stay was conducted among those with COVID-19 hospital admission. Estimates from Model 4, which included: age (continuous term, penalized spline with 6 df) + sex (strata, 2 categories) + smoking status (factor, 3 categories) + individual income (factor, 3 categories) + health risk group (factor, 4 categories) + small area socioeconomic index (continuous term) + percentage of non-Spanish nationals (continuous term) + distance to the closest primary care unit (continuous term) + urbanicity (strata, 3 categories) + average weekly of test-positive proportion (continuous term) + health region (strata, 7 categories).

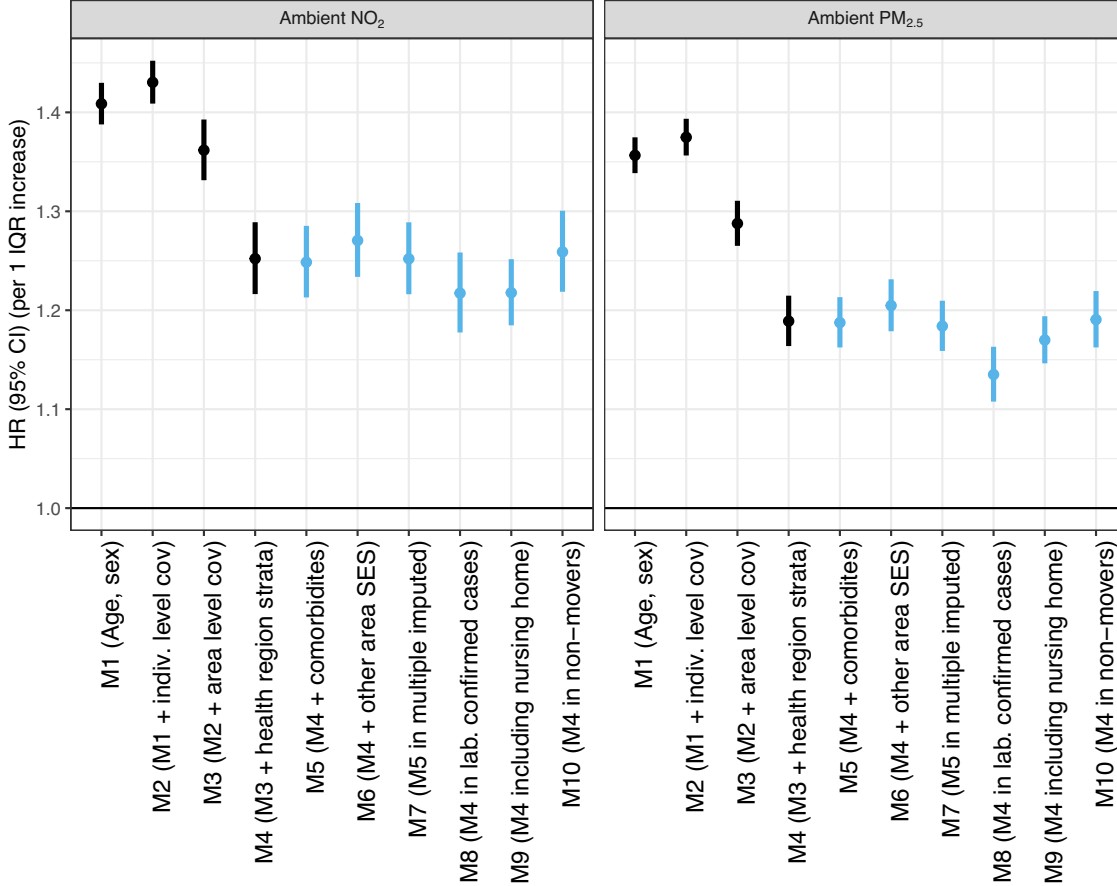

**Fig. 2 | Sequential adjustment and sensitivity analyses for associations between long-term exposure to NO$_2$ and PM$_{2.5}$ and COVID-19-related hospitalization (single-pollutant models).** These estimates are from the sequential adjustment for confounding (black estimates, models 1–4) and six a priori sensitivity analyses (blue estimates, models 5–10), as described in "Methods". Error bars refer to the 95% confidence interval from the Cox Proportional Hazards model. cov denotes covariates; M denotes model; SES denotes socioeconomic status.

There was no clear evidence of departure from linearity for the association between NO$_2$ and PM$_{2.5}$ and COVID-19-related hospitalizations, ICU admissions, and deaths (Supplementary Figs. S5, S6, and S7), particularly in the most common exposure range.

## Discussion

We observed a positive association between long-term exposure to PM$_{2.5}$ and NO$_2$ with severe COVID-19 in this large population-based cohort of adults in Catalonia, Spain, a country with a high burden of COVID-19 in 2020. In sensitivity analyses, associations were stable in two-pollutant models when accounting for different adjustments and when using different outcome definitions and air pollutant exposure models. O$_3$ was positively associated with severe outcomes when adjusted by NO$_2$.

Our estimates for long-term PM$_{2.5}$ and COVID-19-related hospitalization are consistent with other cohorts of COVID-19 cases[9,10]. The association for hospitalization ranged from an odds ratio of 1.06 (95% CI, 1.01–1.12, per 1.7 µg/m³ (IQR) increase) to HR of 1.24 (95% CI,

1.16–1.32, per 1.5 µg/m³ (SD) increase) in analyses conducted in 150,000 COVID-19 cases in Ontario, Canada[10] and 75,000 cases in California, US[9]. In contrast with our findings, analyses in these two cohorts and other individual-level studies[12,15,18] observed no evidence of an association between long-term $NO_2$ and hospitalization. Regarding BC, there was no evidence of an association with severe COVID-19 in two studies that evaluated this pollutant[13,15]. The limited sample size and selected population could explain the differences in our findings.

Overall, our estimates are slightly greater in magnitude than the previous literature for COVID-19 hospitalization (Supplementary Table S15), although a direct comparison is not straightforward because of differences in exposure assessment, confounder adjustment, and outcome definition. One possible explanation for the observed differences is that we analyzed a population-based cohort; thus, our estimates encompassed the risk of infection and the associated risk of severe COVID-19 following infection. In contrast, cohorts including only COVID-19-diagnosed individuals estimated the risk of severe COVID-19 following infection[19]. We observed greater estimates during the first wave, which may reflect higher levels of susceptibility to severe COVID-19 compared to the second wave or unmeasured contextual confounding factors such as spatiotemporal patterns in health system capacity, which were less influential in the second wave.

Estimates for the association of long-term air pollution exposure with COVID-19 death are more inconsistent in the literature compared to those for hospitalization[10–13,16,18]. A population-based cohort study from the general adult population in Rome ($n = 1,594,308$) observed an HR of 1.08 (95% CI, 1.03–1.13, per IQR 0.92 µg/m³ increase) for long-term $PM_{2.5}$ and 1.09 (1.02–1.16, per IQR 9.22 µg/m³ increase) for the long-term $NO_2$ for COVID-19-related deaths[16]. These estimates are smaller than in a population-based cohort of COVID-19 cases ($n = 3,139,804$) in California, US[11], where the estimated long-term $PM_{2.5}$ association with death was a RR of 1.04 (95% CI, 1.03, 1.05), and similar to the estimate in this study (HR of 1.04, 95% CI, 1.02–1.06, both for 1 µg/m³ increase in $PM_{2.5}$). Nevertheless, a cohort with 150,000 COVID-19 cases in Canada reported null associations for death, while positive association for hospital and ICU admission[10]; a cohort of a selected population from the UK (UK-Biobank cohort, $n = 424,721$) observed null results for death for $PM_{2.5}$ (HR 1.00, 95% CI, 0.89–1.11, per IQR 1.27 µg/m³ increase) and $NO_2$ (HR 1.03, 95% CI, 0.90–1.16, per IQR 9.93 µg/m³ increase)[12].

The estimates for the association between $O_3$ and severe COVID-19 are hard to interpret because of its high negative correlation with the other pollutants, particularly $NO_2$ ($r = -0.82$, Supplementary Methods). This could be observed in two-pollutant models with null or positive estimates, contrasting with single-pollutant models.

We evaluated hospital LOS as a surrogate of the COVID-19 severity and burden in the health system[20,21]. The LOS is the result of patient severity, delivered care, and hospital performance, reflecting the required number of staff, beds, and devices and associated costs[20,21]. We observed a positive association between long-term $PM_{2.5}$ and $NO_2$ with hospital LOS. We observed a greater magnitude of the association between ICU admission and hospitalization compared to death, a pattern also observed in the majority of studies that evaluated more than one severity outcome[9,10]. Other individual factors may explain these differences in the magnitude of effect, such as frailty, given that frail individuals were more likely to die out of the hospital or were not eligible for ICU care, especially during the first waves.

There are several biological mechanisms through which long-term air pollution could increase the risk of severe COVID-19. An initial hypothesis was that long-term air pollution increases the baseline risk of the population exposed to higher levels, resulting in a greater prevalence of chronic comorbidities associated with severe COVID-19, such as hypertension. In this case, chronic comorbidities associated with long-term exposure to air pollution would mediate the association between long-term exposure and severe COVID-19. Although we did not perform a formal causal mediation analysis[22], adjustment for chronic comorbidities associated with air pollution in our sensitivity

**Table 3 | Adjusted long-term associations between air pollutants and COVID-19-related outcomes in single-pollutant models by COVID-19 waves**

| Exposure | First wave HR (95% CI) | Second wave HR (95% CI) |
|---|---|---|
| **Hospitalization** | | |
| $NO_2$ (IQR increase: 16.1) | 1.32 (1.27–1.37) | 1.16 (1.11–1.22) |
| $PM_{2.5}$ (IQR increase: 3.2) | 1.25 (1.21–1.28) | 1.11 (1.07–1.14) |
| **ICU admission** | | |
| $NO_2$ (IQR increase: 16.1) | 1.48 (1.32–1.67) | 1.34 (1.18–1.53) |
| $PM_{2.5}$ (IQR increase: 3.2) | 1.19 (1.09–1.30) | 1.12 (1.02–1.23) |
| **Death** | | |
| $NO_2$ (IQR increase: 16.1) | 1.15 (1.06–1.25) | 1.25 (1.10–1.41) |
| $PM_{2.5}$ (IQR increase: 3.2) | 1.12 (1.06–1.20) | 1.14 (1.04–1.25) |
| **Hospital LOS** | | |
| $NO_2$ (IQR increase: 16.1) | 1.06 (1.03–1.10) | 1.03 (0.99–1.06) |
| $PM_{2.5}$ (IQR increase: 3.2) | 1.07 (1.04–1.09) | 1.05 (1.02–1.07) |

Time-stratified Cox model adjusted as Model 4: age (continuous term, penalized spline with 6 df) + sex (strata, 2 categories) + smoking status (factor, 3 categories) + individual income (factor, 3 categories) + health risk group (factor, 4 categories) + small area socioeconomic index (continuous term) + percentage of non-Spanish nationals (continuous term) + distance to the closest primary care unit (continuous term) + urbanicity (strata, 3 categories) + average weekly of test-positive proportion (continuous term) + health region (strata, 7 categories)

**Table 4 | Adjusted long-term associations between air pollutants and COVID-19-related hospitalization in single and two-pollutant models, comparing all-cause with cause-specific hospitalizations**

| | Exposure | All cause ($n = 47,174$) HR (95% CI) | COVID-19 or respiratory* ($n = 36,505$) HR (95% CI) | COVID-19* ($n = 33,981$) HR (95% CI) |
|---|---|---|---|---|
| COVAIR-CAT models | | | | |
| **$NO_2$** (increase: 16.1) | Single-pollutant | 1.25 (1.22–1.29) | 1.27 (1.23–1.32) | 1.27 (1.23–1.32) |
| **$PM_{2.5}$** (increase: 3.2) | Single-pollutant | 1.19 (1.16–1.21) | 1.21 (1.18–1.24) | 1.21 (1.18–1.24) |
| COVAIR-CAT models | | | | |
| **$NO_2$** (increase: 16.1) | Adjusted for $PM_{2.5}$ | 1.12 (1.08–1.17) | 1.13 (1.08–1.19) | 1.12 (1.07–1.18) |
| **$NO_2$** (increase: 16.1) | Adjusted for $O_3$ | 1.24 (1.19–1.29) | 1.29 (1.23–1.35) | 1.26 (1.20–1.32) |
| **$PM_{2.5}$** (increase: 3.2) | Adjusted for $NO_2$ | 1.12 (1.08–1.15) | 1.13 (1.09–1.17) | 1.14 (1.10–1.18) |
| **$PM_{2.5}$** (increase: 3.2) | Adjusted for $O_3$ | 1.16 (1.13–1.19) | 1.19 (1.16–1.22) | 1.18 (1.15–1.22) |

Model adjusted as Model 4: age (continuous term, penalized spline with 6 df) + sex (strata, 2 categories) + smoking status (factor, 3 categories) + individual income (factor, 3 categories) + health risk group (factor, 4 categories) + small area socioeconomic index (continuous term) + percentage of non-Spanish nationals (continuous term) + distance to the closest primary care unit (continuous term) + urbanicity (strata, 3 categories) + average weekly of test-positive proportion (continuous term) + health region (strata, 7 categories).
*Defined by the ICD-10 code first position.

analysis (model 5) resulted in minimal change in the estimates, similar to findings in other cohort studies[11,16], suggesting other direct pathways are more relevant. A limitation to interpreting these estimations is that our main model includes a health risk index, in which chronic comorbidities partially contribute to its estimation[23]. Another hypothesis is that air pollution exposure could facilitate SARS-CoV-2 binding based on evidence that exposure to particulate matter upregulates the expression of SARS-CoV-2 receptors in the lung (e.g., angiotensin-converting enzyme 2)[6]. If this hypothesis is further validated, it is likely the association between air pollution and severe COVID-19 could be driven mainly by other mechanisms than by increasing the overall population risk due to chronic comorbidities. Exposure to air pollution may also be related to changes in immune defenses that are key to mitigating SARS-CoV-2, such as a decrease in type II interferon response to SARS-CoV-2 and antibody response[15,24]. All of these hypothesized mechanisms would result in a population susceptible to severe COVID-19; however, further studies are needed to understand the main biological pathways involved.

Strengths of our analysis include the combination of population representativeness spanning large urban and rural areas, with detailed individual-level data for exposures and confounding adjustment in a country heavily affected by the pandemic during 2020, yielding good statistical power and external validity of our analysis. This allowed us to properly evaluate contrasting results in the literature, such as for $NO_2$ and BC. We evaluated two-pollutant models, a range of complementary outcomes including health system burden, several sensitivity analyses, and assessed the shape of the exposure-response function. Additionally, we used a state-of-the-art exposure assessment model developed for COVAIR-CAT for the study period, providing updated estimates of ambient air pollution in the region at fine spatiotemporal resolution.

We evaluated the first year of the pandemic, a period without COVID-19 vaccines and Variants of Concern; thus, our estimates may not be representative of the effect of air pollution on COVID-19 in the later phases of the pandemic. However, Chen et al. observed positive associations between ambient $PM_{2.5}$ and $NO_2$ and severe COVID-19 after extending the follow-up of an earlier analysis of COVID-19 patients[9] to include the Delta Variant of Concern period[25]. By extending the follow-up, the authors could evaluate the role of vaccination status; initial results showed an association between ambient pollution and severe COVID-19 outcomes in both vaccinated and unvaccinated individuals[25].

We lacked data on some individual-level potential confounders, such as race/ethnicity, migration status, physical activity, and occupation. The adjustment for individual-level income could partially adjust for some of these variables, but residual confounding may still be present.

We operationalized our outcome definition based on a time-defined window from clinically or laboratory-confirmed COVID-19 diagnosis. This allowed us to deal with the lack of access to testing during the first wave and avoid selection bias[26], although some misclassification in COVID-19 diagnosis may have been present for cases not laboratory confirmed. This pragmatic time-defined definition, used in different studies and policy decisions for COVID-19[9,16], captured acute complications of COVID-19 occurring within 30 days of infection but could also include some unrelated COVID-19 hospitalizations. However, results from our sensitivity analyses addressing these limitations, such as analyzing only laboratory-confirmed cases and cause-specific hospitalizations, yielded similar estimates. When evaluating the cohorts of COVID-19 diagnosis in sensitivity analyses, we observed smaller estimates compared with the main analysis; however, estimates based only on individuals who were tested are likely affected by selection bias[27,28].

Long-term exposure to ambient air pollution was positively associated with severe COVID-19 events, including COVID-19-related hospitalization, ICU admission, and deaths, as well as the length of hospital stay in a large, population-based cohort. Our findings add further compelling evidence on the importance of reducing air pollution levels to improve population health generally and severe acute respiratory infection specifically.

## Methods

### Study design and population

We constructed a population-based cohort of the adult population of Catalonia (the northeast region of Spain) as part of the COVAIR-CAT study. The COVAIR-CAT cohort was built through record linkage using data collected in the public health administration databases of Catalonia[22]. The public healthcare system covers nearly the entire population (98.8% of the 7.4 million in 2015)[29]. Catalonia (32,113 km$^2$) is composed of 947 municipalities grouped in seven health regions (median area of 5425 km$^2$). Health regions administer the public health system, accounting for geographical, socioeconomic, demographic, and health facility availability differences, with the aim of guaranteeing equitable healthcare access. Healthcare management areas (AGA, $n = 43$, median area 389 km$^2$) are territorial boundaries based on the aggregation of nested primary care service areas (ABS, $n = 374$, median area 14 km$^2$. Maps of the health areas are shown in Supplementary Methods). These geographic units are used for the operational planning, coordination, and analysis of the main flows between primary care and basic hospital care.

The original cohort included 5,127,059 adults (≥18 years) residents of Catalonia who were covered by the public healthcare system in 2015[22]. COVAIR-CAT includes all individuals from the cohort who were alive and residing in Catalonia on March 1, 2020 ($n = 4,669,011$), excluding the population that arrived in Catalonia between the years 2016 and 2020. We followed participants through December 31, 2020. A detailed description of the cohort construction is in Supplementary Methods.

Data were managed to ensure anonymization in accordance with current data protection legislation by the Agency for Health Quality and Assessment of Catalonia (AQuAS). The cohort design, definitions, and analysis plan were pre-specified in a protocol before any data extraction. Any deviance from the initial plan is labeled as post-hoc. We received approval from our local ethics committee Parc de Salut Mar Ethics Committee (CEIM-PS MAR, no. 2020/9610).

### Data sources

Participants were identified from the Catalan Central Registry of Insured Persons, which collects sociodemographic, migration, and vital status information using a unique identifier. We used this unique identifier for a deterministic linkage across different administrative databases: primary care (CMDB-AP), urgency care (CMDB-URG), and acute hospital discharge (CMBD-AH), which provided detailed information on comorbidity and hospital and ICU admissions based on International Classification of Diseases (ICD) codes (ICD-09 before 2017 and ICD-10 after 2017)[22,30]. Additionally, we used data from a surveillance system of SARS-CoV-2 tests performed in Catalonia (SUVEC) to gather information on RT-qPCR and rapid antigen tests among cohort participants. We used other public sources for area-level covariates, such as the 2011 Spanish Census, satellite data, and a COVID-19 pandemic indicator (i.e., weekly test-positive proportion[31]).

### Outcomes

Our primary outcome was COVID-19-related hospitalization. Secondary outcomes were COVID-19-related death, ICU admission, and hospital length of stay (LOS). We defined a COVID-19-related event as events that occurred within 30 days of COVID diagnosis[9,16]. We defined an individual with a COVID-19 diagnosis as those with a positive RT-qPCR or rapid antigen test (laboratory-confirmed COVID-19 diagnosis) or those with a clinical diagnosis of COVID-19. Clinical diagnosis of COVID-19 was defined by the respective ICD-10 codes, as notified in the administrative healthcare databases. The first COVID-19 diagnosis could be in primary care, urgency care units, or hospitals. We considered COVID-19 diagnoses in the general population, excluding

diagnoses at nursing homes in the main analysis, because of their high frailty, markedly different pattern of COVID-19 spread and eligibility for hospital admission compared to the general population, and their clustered air pollution exposure[10]. For this analysis, we considered only the first COVID-19 diagnosis from March 1, 2020, to December 31, 2020. After identifying the date of the first COVID-19 diagnosis, we defined a COVID-19-related hospitalization as a hospital admission by any cause occurring in the following 30 days and a COVID-19-related death as death by any cause occurring in the following 30 days[16]. To account for individuals who were first hospitalized and had a subsequent COVID-19 diagnosis, particularly during the first wave of the pandemic in Spain, we also considered hospitalizations that occurred in the previous 10 days of the first COVID-19 diagnosis. For each COVID-19-related hospitalization, we retrieved data on whether the participant was admitted to the ICU and the hospital LOS during that hospitalization.

## Exposures

We assessed individual-level exposure to ambient levels of $PM_{2.5}$, $NO_2$, and $O_3$ from the COVAIR-CAT exposure assessment models. We developed an exposure assessment for daily temperature, $PM_{2.5}$, $PM_{10}$, $NO_2$, and maximum 8h-average $O_3$ at a spatial resolution of 250 m for the period 2018–2020 in Catalonia. We used meteorological and air pollution data from the Catalan and Spanish monitoring networks and applied machine learning methods tailored for spatiotemporal prediction (Random Forest-based spatial variable selection)[32]. From the daily estimates, we obtained the annual average of $PM_{2.5}$ and $NO_2$ and the warm season average for $O_3$, corresponding to 2019. The station-based nested 10-fold cross-validation $R^2$ was 0.61 for $PM_{2.5}$, 0.77 for $NO_2$, and 0.87 for $O_3$. In a complementary analysis, we used the annual average estimates of the $PM_{2.5}$, $NO_2$, $O_3$, and BC derived from land-use regression models developed through the ELAPSE (Effects of Low-Level Air Pollution: A Study in Europe) project for 2010[33]. We assigned the 2019 air pollutant exposures to each participant's residential address at the start of 2021 or the last available because we did not have the residential address at the start of 2020 as the address registry for 2020 in Catalonia was disrupted by the pandemic.

Detailed information about the COVAIR-CAT and ELAPSE models is provided in Supplementary Methods.

## Covariates

We obtained age, sex, individual-level income, and health risk group in 2015 from the Central Registry of Insured Persons. Individual income group was based on the co-payment system for drug dispensations, which largely depends on income[22]. Individual health risk group is a validated ordinal index that encompasses multimorbidity and levels of patient complexity, accounting for acute, chronic or oncological morbidities, single or multimorbidity, medications, and demand of the health system[30,34].

Tobacco smoking status (non-smoker, former smoker, or active smoker), previous chronic comorbidities, and body mass index were obtained from the primary care database. Selected chronic comorbidities were also obtained from the hospital admissions database (e.g., chronic obstructive pulmonary disease)[22]. Nursing home status for those with COVID-19 diagnosis was obtained from the COVID-19 surveillance system.

Area-level indicators were linked to individuals' residence addresses. The urbanicity index divided municipalities into towns, urban, and rural areas. The small area socioeconomic index was ascertained at the ABS level[22,35], while the deprivation and Gini indexes and the proportion of non-Spanish residents were ascertained at the census tract level[22]. As a surrogate for public health system accessibility, we derived the distance from the residence to the closest primary care center. Finally, we obtained the weekly test-positive proportion of RT-qPCR and rapid antigen tests at the AGA level.

A detailed description of all covariates is shown in Supplementary Methods.

## Data analysis

We described continuous variables using mean ± standard deviation (SD) or median [p25–75] and categorical variables as proportions. There were missing values for tobacco smoking and body mass index covariates. For the main analysis, we considered a missing value on tobacco smoking as a non-smoker because the value is most often omitted for non-smokers in the primary care service, while body mass index was used only for sensitivity analysis after multiple imputations.

We fit Cox proportional hazards models to estimate the association between the 2019 annual average air pollution and COVID-19-related hospitalization, ICU admission, and death, with separate models for each pollutant and outcome. The analyses of COVID-19-related hospitalization, ICU admission, and death were conducted in the whole population[19], while the analysis of hospital length of stay was conducted on those individuals with COVID-19-related hospitalization. Our main analyses are based on the COVAIR-CAT estimates for 2019, and we evaluated single- and two-pollutant models. We accounted for the competing event of death when evaluating COVID-19-related hospitalization and ICU admission by censoring a death event using the cause-specific HR framework[36,37]. Follow-up started on March 1, 2020, and for the primary outcome (COVID-19 hospitalization), right-censoring occurred at the first instance of death, 30 days after the first COVID-19 diagnosis, emigration outside the study area, or the end of the study period. We used the time from March 1, 2020, in days as the time scale in the time-to-event analysis. We assessed the proportional hazards assumption of our models by visual inspection of score residuals plotted against event time. We fitted negative binomial regression models to estimate the association between the 2019 annual average air pollution and hospital LOS among those individuals that were hospitalized[38]. Measures of association for air pollutants were reported as hazard ratios (HR) or incidence rate ratios (IRR) per interquartile range (IQR) increase, with their 95% confidence intervals (CI).

We performed the following sequential adjustment for all exposures and outcomes, as pre-defined based on a priori theoretical assumptions about the relationship between the covariates and the outcome:

a. Model 1, adjusted for age (fitted as a penalized spline with six degrees of freedom, number of degrees of freedom evaluated by AIC value) and sex (strata, 2 levels);
b. Model 2, Model 1 plus tobacco smoking status (factor, 3 categories), individual income (factor, 3 categories), and health risk group (factor, 4 categories);
c. Model 3, Model 2 plus area-level covariates: small area socioeconomic index (continuous term), the proportion of non-Spanish nationals (continuous term), distance to the closest primary care unit (continuous term) + urbanicity (strata, 3 categories) and average weekly of test-positive proportion (continuous term); and
d. Model 4 (main model), Model 3 plus health region (strata, 7 categories).

We performed six sensitivity analyses defined a priori:

a. Model 5 included potential mediators (diabetes, chronic obstructive pulmonary disease, obesity, dyslipidemia, hypertension, and other cardiovascular disorders) to Model 4;
b. Model 6 included other socioeconomic indexes (inequity index, Gini, and deprivation index) to Model 4;
c. Model 7 included multiple imputations with chained equations to impute tobacco smoking status and body mass index, running Model 5 and replacing obesity by body mass index in 10 imputed datasets;
d. Model 8 included Model 4, with the outcomes restricted to laboratory-confirmed COVID-19;

e.   Model 9 was Model 4 but included COVID-19 diagnoses at nursing homes, and

f.   Model 10 included Model 4 in the subpopulation that did not move ABS between 2015 and 2020.

Additional ad hoc sensitivity analyses based on our main model (Model 4) included: (1) censoring the cohort on December 1, 2020, allowing a maximum of 30 days to occur the event during the follow-up; (2) adding distance to the nearest hospital; (3) adding population density at the census tract level; (4) adjusting the smoking status by using a missing indicator instead of considering the missing as "never" smokers; (5) running the analysis on the cohort with COVID-19 diagnosis; (6) running the analysis on the cohort with COVID-19 diagnosis at the primary care; (7) considering hospitalizations with COVID-19 as the main cause of admission instead of all-cause admissions.

We evaluated the potential nonlinearity for age and body mass index testing three to six degrees of freedom in a penalized spline. We compared the AIC criteria for each model to determine whether nonlinearity was present.

We conducted several complementary analyses. To explore the exposure modeling assessment, we replicated all previous models (Model 1 to Model 10) using $PM_{2.5}$, $NO_2$, $O_3$, and BC from the ELAPSE model[33] and Model 4 (main analysis) using the COVAIR-CAT estimates for 2018. We explored potential effect modification by the first and second COVID-19 waves in Catalonia. We fit a time-stratified Cox proportional hazards model defining strata by Wave 1 (March 1 to June 20, 2020) and Wave 2 (June 21, 2020, to December 31, 2020) in Model 4 of the main analysis. The periods defining waves were defined by splitting the study period into the week with the lowest number of COVID-19 cases in Catalonia (Fig. 1). To explore the definition of COVID-19-related hospitalization, we fit the Model 4 for COVID-19-related hospitalization considering only admissions with COVID-19 or respiratory as the main cause of admission instead of all-cause admissions.

Finally, to explore potential non-linear exposure-response functions between air pollutants and outcomes, we fit Model 4 of the main analysis but allowing for nonlinearity using penalized splines with three degrees of freedom. A detailed description of all models, complementary analyses, and multiple imputations is provided in Supplementary Methods.

All analyses were conducted in R (R Core Team, 2020) software (version 4.1.2).

### Reporting summary
Further information on research design is available in the Nature Portfolio Reporting Summary linked to this article.

## Data availability
In accordance with current European and national law, the data used in this study is only available for the researchers participating in this study. Thus, we are not allowed to distribute or make the data publicly available to other parties. Researchers can request data from the Agency for Health Quality and Assessment of Catalonia (AQuAS) by contacting the Àrea Programa d'Analítica de Dades per a la Recerca i la Innovació en Salut (PADRIS, padris@gencat.cat). Further information on the requirements and how to access the data are available at https://aquas.gencat.cat/ca/fem/intelligencia-analitica/padris/index.html#-googtrans(ca|en), including the COVID-19 prioritization procedure https://aquas.gencat.cat/ca/fem/intelligencia-analitica/padris/procedi-ment-urgent-prioritzacio-proposes-estudis-sarscov2-covid-19/index.html#googtrans(ca|en).

## Code availability
This study did not generate new or customized code/algorithm. The Cox Proportional models were fit using the function coxph from the

survival R package[39]. The codes used in the analysis are available from the corresponding author.

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

## Acknowledgements
We acknowledge ELAPSE team for the 2010 exposure estimates for Catalonia. This work was supported by the Health Effects Institute (HEI) research agreement (grant No 4980-RFA20-1B/21-3). The research described in this article was conducted under contract with the HEI, an organization jointly funded by the US Environmental Protection Agency (EPA) (assistance award No R-82811201) and certain motor vehicle and engine manufacturers. The contents of this article do not necessarily reflect the views of the HEI or its sponsors, nor do they necessarily reflect the views and policies of the EPA or motor vehicle and engine manufacturers. O.T.R. is supported by a Sara Borrell fellowship from the Instituto de Salud Carlos III (CD19/00110). T.D.S. acknowledges receiving financial support from the Instituto de Salud Carlos III (ISCIII; Miguel Servet 2021: CP21/00023), which is co-funded by the European Union. We acknowledge support from the Spanish Ministry of Science and Innovation and State Research Agency through the "Centro de Excelencia Severo Ochoa 2019–2023" Program (CEX2018-000806-S) and from the Generalitat de Catalunya through the CERCA Program.

## Author contributions
O.T.R. and C.T. conceived and designed the study. O.T.R., A.R., S.O., and C.T. designed and extracted the data. C.M. and C.T. designed and performed the exposure model assessment. O.T.R. performed the statistical analysis and wrote the original draft of the manuscript. A.A., S.O., C.M., A.R., J.B., X.B., C.C., P.D., T.D., M.F., M.N., J.S., A.V., M.K., U.L., C.A., R.V., and C.T. reviewed the manuscript and edited the original version. All authors have read and revised the manuscript for important intellectual content and contributed to the interpretation of the results. All authors have approved the final draft of the manuscript. C.T. is the study guarantor. The corresponding author attests that all listed authors meet authorship criteria and that no others meeting the criteria have been omitted.

## Competing interests
The authors declare no competing interests.
