## [Peer Review File · Nature Communications]

Long-term exposure to air pollution and severe COVID-19 in Catalonia: a population-based cohort studyREVIEWER COMMENTS

Reviewer #1 (Remarks to the Author):

Thank you for the opportunity to review this manuscript on long-term air pollution exposure and COVID-19 severity.

The purpose of the current study was to assess associations of particulate matter, nitrogen dioxide, ozone, and black carbon on COVID-19 severity (hospitalizations, ICU admissions, and death). The manuscript has great potential to add to the literature on environmental exposures and COVID-19 and provides novel findings in a large cohort setting. The analysis is well executed, but requires some additional details to make it reproducible. Most of my comments are minor (below), however, please carefully consider point 6, 7, and 14, which may require rerunning some analyses. Please also note that the following comments are not intended to degrade the researchers but are provided to improve the quality of this work for future submission.

1. Black carbon and hospital length of stay should be mentioned in abstract if they are assessed in the main manuscript.
2. For readers unfamiliar with Catalonia, it may be helpful to have a little more relevant information on the general population (e.g. % insured, % covered versus % not covered by the public healthcare system in 2015).
3. Please add a few lines on the proposed mechanism/rational for study inclusion for each of the three pollutants in the introduction.
4. Are the COVAIR-CAT exposure assessment models published or validated elsewhere? If so, please add references after the phrase, "using machine learning methods". If not, perhaps a couple of sentences could be added to the methods to briefly describe the exposure assessment (spatial resolution, inputs, etc.).
5. It is not clear why the air pollution was assigned to addresses after follow-up (2021) rather than before, was 2019 address information unavailable? Please add either rationale or limitation to the main text.
6. My main concern is that the term COVID-19 hospitalization is misleading. All cause hospitalization following a COVID-19 infection is not comparable with COVID-19 hospitalization, with the latter being used in almost all of the cited literature. Most studies use COVID-19 ICD-10 in a primary position at diagnosis. For comparability with other cited studies, the definition of COVID-19 hospitalization should be "COVID-19 as main cause of admission 33,981 (72.0%)", from Table S3. Table S4 groups respiratory cause admissions, including chronic respiratory, with COVID-19 admissions, but it is important that COVID-19 hospitalizations are reported separately from other respiratory disease admissions. That is, for comparability with other literature, COVID-19 hospitalizations should be the main analysis that is compared to other studies in the discussion.
7. L180 "For the main analysis, we considered a missing value on tobacco smoking as non-smoker because the value is most often omitted for non-smokers in the primary care service", requires a reference to support this decision. Otherwise, to avoid exclusions or imputation, perhaps missing smoker could be coded using a missing indicator e.g. "never", "ever", "smoker", "missing". See <https://arxiv.org/abs/2111.00138> for more information on why I suggest this approach (no need to cite).
8. Please consider if time-to-event is relevant for long-term time invariant exposure (air pollution) and a short follow-up period? Is the time from March 1st 2020 until COVID diagnosis meaningful?
9. When comparing results with other studies please report if they reported HR or OR, etc. before each estimate.
10. L373 "Although we did not perform a formal causal mediation analysis because of the lack of established methods for time to event analysis". Causal mediation is possible with Cox PH, e.g. <https://www.ncbi.nlm.nih.gov/pmc/articles/PMC5010419/>, please correct this statement.
11. This study reports effect estimates that are substantially larger than cited studies with adequate confounder adjustment. Given the infectious nature of COVID-19, perhaps the categorical Urbanicity variable does not fully adjust for the confounding by population density (and perhaps housing conditions). If you agree, this consideration could be discussed in limitations or be worked into the interpretation in the discussion.
12. How was 6 degrees of freedom selected for the age spline?
13. Please comment on the protective effect of O3 in single pollutant models.

14. Finally, in the manuscript you suggest that Figure 3. shows a departure from linearity, but the findings are reported per IQR increase in exposure (linearly). Please consider this point and make the required changes (either to the analysis or to the text). Personally, I don't think that these curves look particularly non-linear in the area where most of the participants are, but given the large sample size you may disagree. Figure 3 could go in the supplement if the phrase about the nonlinear relationship is removed from the main text.

15. L68 please correct "dioxide nitrogen" to "nitrogen dioxide".

16. L236 to help the reader, I suggest a change to, "The annual average levels of air pollution in the cohort were $13.9 \pm 2.2 \mu\text{g}/\text{m}^3$ for PM_{2.5}, $26.2 \pm 10.3 \mu\text{g}/\text{m}^3$ 263 for NO₂ and $91.6 \pm 8.2 \mu\text{g}/\text{m}^3$ 264 for O₃ from the COVAIR-CAT 2019 models".

17. L259 consider changing the paragraph structure so that it is clear to the reader what the denominator is in this sentence.

Reviewer #2 (Remarks to the Author):

The authors investigated the relationship between long-term air pollution and a few COVID-19-related health outcomes in 4.6 million adults from Catalonia, Spain in 2020. The findings are interesting and fill in a research gap on the effect of ambient air pollution in individuals with positive COVID-19 tests. I have a few concerns about the exposure assessment and target sample selection for statistical models. My specific comments are listed below:

Major comment:

Page 6, lines 131-132: since the participants were followed through December 31st, 2020, aren't the authors supposed to define the first COVID-19 diagnosis from March 1st, 2020 to December 1st, 2020 (instead of December 31st) to allow for sufficient time to observe the health outcomes, given that the time range of COVID-19 related events after testing positive is 30 days.

Page 7, lines 146-147, was the annual average exposure computed as the average from Jan 1, 2019 to Dec 31, 2019? If so, this exposure measurement may not be ideal since the date of the first positive test can be any time in 2020, and the discrepancy may cause exposure misclassification. A more appropriate approach is to estimate long-term air pollution as a 365-day average prior to the date of the first positive test.

The cross-validation R₂s were quite low for PM_{2.5} and NO₂ (especially PM_{2.5}). There are other well-validated open data sources for ambient PM_{2.5} and NO₂ that may achieve higher accuracy, see the reference below:

- Aaron van Donkelaar, Melanie S. Hammer, Liam Bindle, Michael Brauer, Jeffery R. Brook, Michael J. Garay, N. Christina Hsu, Olga V. Kalashnikova, Ralph A. Kahn, Colin Lee, Robert C. Levy, Alexei Lyapustin, Andrew M. Sayer and Randall V. Martin (2021). Monthly Global Estimates of Fine Particulate Matter and Their Uncertainty Environmental Science & Technology. doi:10.1021/acs.est.1c05309.

- Cooper, M.J., Martin, R.V., Hammer, M.S., et al. (2022) Global fine-scale changes in ambient NO₂ during COVID-19 lockdowns. Nature 601, 380–387. DOI: 10.1038/s41586-021-04229-0

Page 7, lines 153-154, why did the authors assign the air pollutants at the start of 2021 or the last available? Is this a misspelling? Ideally, the air pollutants should be measured prior to the date of the first positive test (that should be in the middle of 2020, not 2021).

Page 9, Lines 195-197, why would the authors use negative binomial regressions for LOS? negative binomial models are commonly used when the dependent variable is a count variable, and LOS does not look like a count variable to me.

The authors did not mention the sample for each outcome. My understanding is that the authors conducted all the models based on all the 4660502 participants included in the study (please correct me if I'm wrong). If so, those who were never diagnosed with COVID-19 should not be in the risk set. Because they are COVID-19 negative, their risks of COVID-19-related hospitalization,

ICU admission, LOS, and death are 0, regardless of their air pollution levels, demographic characteristics, or others. When conducting survival analyses, the models should be conducted among participants who are at risk at baseline. When the outcomes are COVID-19-related hospitalization, LOS, and death, the study sample should be those who tested positive; when the outcome is ICU admission, the sample should be those who were hospitalized due to COVID-19 (assuming that ICU admission were all transferred from regular admissions).

The estimates for O₃ are confusing and may be misleading. It is hard to interpret O₃ as some sort of "good air". What are the correlation coefficients between PM_{2.5}, NO₂, and O₃? I'm not sure if it is a good idea to put the results of O₃ in a paper that is primarily targeting PM_{2.5} and NO₂.

Minor comments:

Page 6, lines 125-128, the definition of a clinical diagnosis of COVID-19 is not clear to me. Could the authors provide more details about this?

Page 8, lines 163-166, what is the difference between previous chronic comorbidities and individual health risk groups? They were both based on the individual's chronic comorbidities. Claiming acute myocardial infarction as a chronic comorbidity does not make sense to me (acute is an antonym to chronic).

Is it possible to tell different COVID variants for the 340,608 COVID-19-positive individuals in the analysis?

Page 15, line 321: please refrain from using "RR" for logistic regression or generalized linear models, they are not the same thing (even though they are close in certain situations).

Page 19 408-409: what is the general composition of race/ethnicity? and what are the potential impacts on the findings of this study?

Reviewer #3 (Remarks to the Author):

The paper is very well-written. Some key strengths of this study include its use of state-of-the-art methods for air pollution exposure assessment and the population-based cohort study design that consisted of nearly entire adult population in Catalonia, Spain. The findings suggest that long-term exposures to PM_{2.5} and NO₂ were associated with severe COVID-19 outcomes in this cohort. The evidence concerning O₃ exposure is less clear. The authors are commended for conducting a number of sensitivity analyses that resulted in similar findings indicating the robustness of the results. Although several individual-level factors such as race/ethnicity and occupation could not be controlled for, the set of other covariates was ample.

I only have few minor comments and suggestions that are listed below.

Methods/Covariates

1. For many readers who are unfamiliar with Catalonia, please provide more details about this study area, especially the size of various geographic regions used in this study (eg, health regions and primary care service areas). It would helpful to also discuss the appropriateness of adjusting for variables derived at these geographic levels.
2. Could the minimal impact of further adjusting for preexisting comorbidities on the air pollution-COVID-19 associations be due to the fact that the main model already included health risk group? As it was described, the latter variable encompassed multimorbidity. Please clarify.
3. As the primary outcome of interest is COVID-19 hospitalizations, could the authors consider a sensitivity analysis to adjust for distance to a nearest acute-care hospital?
4. Page 9. The authors "accounted for the competing event of death when evaluating COVID-19 related hospitalization and ICU admission by censoring at a death event". Since ambient air pollution has been linked to various mortality outcomes, the censoring may introduce selection

bias under the null. Could the authors consider a sensitivity analysis to account for the censoring using the measured risk factors for the outcome of interest (for example, by IP censoring weighting)?

Discussion

5. Page 18. The authors discussed "Another hypothesis is that air pollution exposure could facilitate SARS-CoV2 binding, because there is evidence that exposure to particulate matter upregulates the expression of SARS-CoV-2 receptors in the lung (e.g., angiotensin-converting enzyme 2)". This manuscript can benefit from a more thorough discussion about this possible mechanism, since its results suggested that the impact of air pollution on increasing the prevalence of chronic comorbidities associated with severe COVID-19 was likely minimal. This will help improve the biological plausibility of the study results.

REVIEWER COMMENTS

Reviewer #1 (Remarks to the Author):

Thank you for the opportunity to review this manuscript on long-term air pollution exposure and COVID-19 severity.

The purpose of the current study was to assess associations of particulate matter, nitrogen dioxide, ozone, and black carbon on COVID-19 severity (hospitalizations, ICU admissions, and death). The manuscript has great potential to add to the literature on environmental exposures and COVID-19 and provides novel findings in a large cohort setting. The analysis is well executed, but requires some additional details to make it reproducible. Most of my comments are minor (below), however, please carefully consider point 6, 7, and 14, which may require rerunning some analyses. Please also note that the following comments are not intended to degrade the researchers but are provided to improve the quality of this work for future submission.

ANSWER: Thank you for your comments.

1. Black carbon and hospital length of stay should be mentioned in abstract if they are assessed in the main manuscript.

ANSWER: We added black carbon and hospital length of stay results to the abstract.

2. For readers unfamiliar with Catalonia, it may be helpful to have a little more relevant information on the general population (e.g. % insured, % covered versus % not covered by the public healthcare system in 2015).

ANSWER: We agree and added this information in the methods about the general population in Catalonia in 2015. In 2015, 98.8% of the total population was covered by the public healthcare system.

3. Please add a few lines on the proposed mechanism/rational for study inclusion for each of the three pollutants in the introduction.

ANSWER: We added the mechanistic rationale for analysing them in the introduction.

4. Are the COVAIR-CAT exposure assessment models published or validated elsewhere? If so, please add references after the phrase, “using machine learning methods”. If not, perhaps a couple of sentences could be added to the methods to briefly describe the exposure assessment (spatial resolution, inputs, etc.).

ANSWER: The COVAIR-CAT exposure modeling manuscript is about to be submitted for publication. We improved the description of the modeling approach in the methods. The original submission included a description of the models in the supplementary methods.

5. It is not clear why the air pollution was assigned to addresses after follow-up (2021) rather than before, was 2019 address information unavailable? Please add either rationale or limitation to the main text.

ANSWER: We agree this was not sufficiently clear. We used the residential address of each individual at the start of 2021 or the last available, because we did not have an updated residential address for the start of 2020. The address registry for 2020 in Catalonia was disrupted by the pandemic. We had access to residential addresses at the start of 2015 or 2021. We selected the start of 2021, as it was closer to 2020 and run a sensitivity analysis accounting for those that did not move between 2015 and January 2021. We have now clarified the text.

6. My main concern is that the term COVID-19 hospitalization is misleading. All cause hospitalization following a COVID-19 infection is not comparable with COVID-19 hospitalization, with the latter being used in almost all of the cited literature. Most studies use COVID-19 ICD-10 in a primary position at diagnosis. For comparability with other cited studies, the definition of COVID-19 hospitalization should be “COVID-19 as main cause of admission 33,981 (72.0%)”, from Table S3. Table S4 groups respiratory cause admissions, including chronic respiratory, with COVID-19 admissions, but it is important that COVID-19 hospitalizations are reported separately from other respiratory disease admissions. That is, for comparability with other literature, COVID-19 hospitalizations should be the main analysis that is compared to other studies in the discussion.

ANSWER: Thanks for the opportunity to clarify this point. We have discussed this approach during the protocol planning. We chose the time-defined window, ie, any hospitalization within 30 days of COVID-19 diagnosis, because this definition has been used since the beginning of the pandemic in clinical trials for treatment,¹⁻³ as well as, by national public health agencies (e.g., UK)⁴ and other environmental epidemiological studies.^{5,6} One of the main advantages of this approach is to avoid the problem of ICD-10 coding practice that can vary within and between hospitals; this is particularly important during the first wave. Being a new disease in 2020, hospitalizations that today would be coded as due to COVID-19 were not consistently coded in the first wave due to lack of knowledge and standardized coding (e.g., an acute respiratory failure with “happy or silent hypoxaemia”⁷ clinical presentation - severe hypoxemia and absence of dyspnoea - or a stroke in a young patient infected SARS-CoV-2 patients). For instance, we have 2,524 hospital admissions as Respiratory cause as main cause (first position of ICD-10 code), in individuals with a recent COVID-19 diagnosis. Additionally, we did not analyse the Omicron period, when incidental hospitalizations could occur, thus decreasing the likelihood of COVID-19 not being responsible for that hospitalization.

We agree with the reviewer that this is an important topic, and now also show in the Main text (New Table 4) the results of the new analysis requested, using the main model adjustment and having COVID-19 as the main/first cause of hospitalization. The results are consistent to our primary analysis and add robustness to our manuscript. However, we did not change the time-window definition in our main analysis in order to remain consistent with our original study protocol.

7. L180 “For the main analysis, we considered a missing value on tobacco smoking as non-smoker because the value is most often omitted for non-smokers in the primary care service”, requires a reference to support this decision. Otherwise, to avoid exclusions or imputation, perhaps missing smoker could be coded using a missing indicator e.g. “never”, “ever”, “smoker”, “missing”. See <https://arxiv.org/abs/2111.00138> for more information on why I suggest this approach (no need to cite).

ANSWER: We discussed this topic during the development of our analysis plan and based our approach on recommendations from agencies responsible for the collection and analysis of administrative health data in Catalonia. Unfortunately, we did not find any data or references that directly support this approach. Nevertheless, our original submissions included a sensitivity analysis using multiple imputation, showing similar results. In the revised version, we run an additional sensitivity analysis using the missing indicator and show it in the supplement (**supplementary tables S3, S4, S5**); again, with results consistent with our main analysis. While the reference mentioned by the reviewer supports the use of a missing indicator, others references highlight its potential problems.⁸

		Smoking missing as Never Smoker (main analysis)	Smoking missing after Multiple Imputation	Smoking missing as Missing Indicator
	Exposure	HR (95% CI)	HR (95% CI)	HR (95% CI)
NO₂ (IQR increase: 16.1)	Single-pollutant	1.25 (1.22-1.29)	1.25 (1.22-1.29)	1.26 (1.22-1.29)
PM_{2,5} (IQR increase: 3.2)	Single-pollutant	1.19 (1.16-1.21)	1.18 (1.16-1.21)	1.18 (1.16-1.21)
O₃ (warm season) (IQR increase: 10.8)	Single-pollutant	0.91 (0.89-0.92)	0.90 (0.89-0.92)	0.90 (0.89-0.92)

8. Please consider if time-to-event is relevant for long-term time invariant exposure (air pollution) and a short follow-up period? Is the time from March 1st 2020 until COVID diagnosis meaningful?

ANSWER: Thanks for this comment. Adjusting for calendar/epidemic time is important to capture the dynamics of the epidemic in terms of temporal changes in transmissibility, diagnosis, treatment, etc,. Thus, we used a time-to-event analysis with daily time steps because of the advantages of accounting for censoring and time trends in the baseline hazard due to the dynamics of the epidemic.

9. When comparing results with other studies please report if they reported HR or OR, etc. before each estimate.

ANSWER: Thanks for this. We revised the discussion and added the measured estimate for all values.

10. L373 “Although we did not perform a formal causal mediation analysis because of the lack of established methods for time to event analysis”. Causal mediation is possible with Cox PH, e.g. <https://www.ncbi.nlm.nih.gov/pmc/articles/PMC5010419/>, please correct this statement.

ANSWER: The reviewer is correct, and we deleted that sentence.

11. This study reports effect estimates that are substantially larger than cited studies with adequate confounder adjustment. Given the infectious nature of COVID-19, perhaps the categorical Urbanicity variable does not fully adjust for the confounding by population density (and perhaps housing conditions). If you agree, this consideration could be discussed in limitations or be worked into the interpretation in the discussion.

ANSWER: We run a sensitivity analysis with population density at the census tract level, now reported in **supplementary tables S3, S4, S5**. The results are robust to this further adjustment.

12. How was 6 degrees of freedom selected for the age spline?

ANSWER: We evaluated from 3 to 6 degrees of freedom and chose the best fit by using AIC criteria. We clarified this in the revised version.

13. Please comment on the protective effect of O₃ in single pollutant models.

ANSWER: Thanks for this comment. Following the editor and reviewers' comments, we present all O₃ results in the supplement. We agreed it is difficult to interpret the O₃ coefficients given the high negative correlation with the other pollutants. We highlight the issue of potential confounding by correlated co-pollutants in the revised version of the manuscript, particularly in the interpretation of changes in the O₃ estimates when co-adjusting by NO₂.

The correlation between the air pollutants is in the supplementary appendix and shown below.

PM_{2.5} x NO₂: 0.89

PM_{2.5} x O₃: -0.76

NO₂ x O₃: -0.82

14. Finally, in the manuscript you suggest that Figure 3. shows a departure from linearity, but the findings are reported per IQR increase in exposure (linearly). Please consider this point and make the required changes (either to the analysis or to the

text). Personally, I don't think that these curves look particularly non-linear in the area where most of the participants are, but given the large sample size you may disagree. Figure 3 could go in the supplement if the phrase about the nonlinear relationship is removed from the main text.

ANSWER: We agree with the reviewer. We reduced the emphasis on the non-linearity of the association and moved these results to the supplement.

15. L68 please correct “dioxide nitrogen” to “nitrogen dioxide”.

ANSWER: We corrected it.

16. L236 to help the reader, I suggest a change to, “The annual average levels of air pollution in the cohort were $13.9 \pm 2.2 \mu\text{g}/\text{m}^3$ for PM_{2.5}, $26.2 \pm 10.3 \mu\text{g}/\text{m}^3$ 263 for NO₂ and $91.6 \pm 8.2 \mu\text{g}/\text{m}^3$ 264 for O₃ from the COVAIR-CAT 2019 models”.

ANSWER: Thanks. We changed it.

17. L259 consider changing the paragraph structure so that it is clear to the reader what the denominator is in this sentence.

ANSWER: We changed the paragraph clearly specifying the denominators. Thanks.

Reviewer #2 (Remarks to the Author):

The authors investigated the relationship between long-term air pollution and a few COVID-19-related health outcomes in 4.6 million adults from Catalonia, Spain in 2020. The findings are interesting and fill in a research gap on the effect of ambient air pollution in individuals with positive COVID-19 tests. I have a few concerns about the exposure assessment and target sample selection for statistical models. My specific comments are listed below:

ANSWER: Thank you for your comments.

Major comment:

Page 6, lines 131-132: since the participants were followed through December 31st, 2020, aren't the authors supposed to define the first COVID-19 diagnosis from March 1st, 2020 to December 1st, 2020 (instead of December 31st) to allow for sufficient time to observe the health outcomes, given that the time range of COVID-19 related events after testing positive is 30 days.

ANSWER: Thanks for this comment. We now explore this point in a sensitivity analysis. The majority of hospital admissions occurred very shortly after the COVID-19 diagnosis: mean

3.8 ± 5 days, 83% occurred in the first week after diagnosis and 95% within 2 weeks. We therefore did not anticipate a large impact of using the 31st Dec as the administrative censoring date and preferred to keep the whole year to include more COVID-19 diagnoses. In the sensitivity analysis, the results are robust to changing the censoring date (supplementary tables S3, S4, S5).

Page 7, lines 146-147, was the annual average exposure computed as the average from Jan 1, 2019 to Dec 31, 2019? If so, this exposure measurement may not be ideal since the date of the first positive test can be any time in 2020, and the discrepancy may cause exposure misclassification. A more appropriate approach is to estimate long-term air pollution as a 365-day average prior to the date of the first positive test.

ANSWER: Thanks for highlighting this point. We decided to use the 2019 average because 1) for individuals without COVID-19 diagnosis, we don't have a diagnosis/test date to define the 365-day average; 2) this approach could add seasonal variation in the exposure and open the analysis to additional risk of residual confounding by seasonally varying factors; and 3) annual averages across years and with a sample of 365 running averages are highly correlated as shown below. Therefore, based on their high correlation, we believe that misclassification is unlikely.

Annual Averages Correlations (Spearman correlation)

PM_{2.5}

	2018	2019	2020
2018	1		
2019	0.979	1	
2020	0.972	0.958	1

NO₂

	2018	2019	2020
2018	1		
2019	0.991	1	
2020	0.985	0.997	1

O₃

	2018	2019	2020
2018	1		
2019	0.965	1	
2020	0.869	0.893	1

Additionally, as suggested by the reviewer, we sampled 500 points in a regular grid within the study area, extracted the exposure values, computed 1-year rolling means at 12 time points (every 1st of the month from Jan to Dec 2019) and computed the Pearson correlation between them. In the correlation plots, the date indicates the first day of the rolling average: 2019-01-01 means our current exposure average (2019-01-01 to 2019-12-31); someone with a COVID-19 diagnosis on 2020-04-01 has its previous 365 average marked as 2019-04-01 in the plots, meaning average between 2019-04-01 and 2020-03-31

PM_{2.5}

NO₂

The cross-validation R2s were quite low for PM2.5 and NO2 (especially PM2.5). There are other well-validated open data sources for ambient PM2.5 and NO2 that may achieve higher accuracy, see the reference below:

- Aaron van Donkelaar, Melanie S. Hammer, Liam Bindle, Michael Brauer, Jeffery R. Brook, Michael J. Garay, N. Christina Hsu, Olga V. Kalashnikova, Ralph A. Kahn, Colin Lee, Robert C. Levy, Alexei Lyapustin, Andrew M. Sayer and Randall V. Martin (2021). **Monthly Global Estimates of Fine Particulate Matter and Their Uncertainty** *Environmental Science & Technology*. doi:10.1021/acs.est.1c05309.
- Cooper, M.J., Martin, R.V., Hammer, M.S., et al. (2022) **Global fine-scale changes in ambient NO2 during COVID-19 lockdowns.** *Nature* 601, 380–387. DOI: 10.1038/s41586-021-04229-0

ANSWER: Thanks for these suggestions. We agree the performance statistics for PM_{2.5} models were not ideal and reflected the characteristics of the PM_{2.5} monitoring network in Catalonia. More than 80% of the PM_{2.5} observations in the region come from manual stations, which typically record daily average PM_{2.5} with frequency lower than daily. This makes their use for exposure modeling challenging, since they do not have enough temporal coverage to model long-term concentrations. Even though we use the best from the data by applying advanced modeling, which borrowed information from PM₁₀. As a sensitivity analysis we have the analysis with ELAPSE PM_{2.5} models, yielding similar results. The reference suggested are monthly global estimates with an spatial resolution of 0.1° x 0.1° degrees (~ 11km), while our modeling was customized to the region and had a resolution of 250 meters.

Regarding NO₂, the exposure metrics the reviewer suggested are mostly based on 5P TROPOMI tropospheric NO₂ columns (which our model also includes) at 1km resolution. Our model incorporates many other local relevant predictors (e.g. road network, land cover, population density, impervious surfaces) and offers better spatial resolution (250m) than the suggested exposure estimates, which is critical for long-term estimation of NO₂, which may vary a lot at short distances. Nonetheless and similarly to PM_{2.5}, we also linked ELAPSE NO₂ estimates (100 m resolution, year 2010), which were highly correlated with COVAIR exposures and achieved similar results in epidemiological models.

Page 7, lines 153-154, why did the authors assign the air pollutants at the start of 2021 or the last available? Is this a misspelling? Ideally, the air pollutants should be measured prior to the date of the first positive test (that should be in the middle of 2020, not 2021).

ANSWER: The long-term exposure to the air pollutants were from 2019, ie, previous to the pandemic, to characterize the long-term exposure. We agree this was not sufficiently clear. We used the residential address of each individual at the start of 2021 or the last available, because we did not have an updated residential address for the start of 2020. The address registry for 2020 in Catalonia was disrupted by the pandemic. We had access to residential addresses at the start of 2015 or 2021. We selected the start of 2021, as it was closer to 2020 and run a sensitivity analysis accounting for those that did not move between 2015 and January 2021. We have now clarified the text.

Page 9, Lines 195-197, why would the authors use negative binomial regressions for LOS? negative binomial models are commonly used when the dependent variable is a count variable, and LOS does not look like a count variable to me.

ANSWER: Thanks for the opportunity to clarify this point. Length-of-stay is usually a count variable (1, 2, 3, 4, 5 days spent in the hospital). In our analysis, we generated counts of days in hospital based on date of admission and date of discharge. We recognize that there is not a consensus in the literature on how to model LOS; however, negative binomial is one of the most commonly used modelling approach in several benchmark papers.⁹⁻¹¹ We have referenced it in the manuscript.

The authors did not mention the sample for each outcome. My understanding is that the authors conducted all the models based on all the 4660502 participants included in the study (please correct me if I'm wrong). If so, those who were never diagnosed with COVID-19 should not be in the risk set. Because they are COVID-19 negative, their risks of COVID-19-related hospitalization, ICU admission, LOS, and death are 0, regardless of their air pollution levels, demographic characteristics, or others. When conducting survival analyses, the models should be conducted among participants who are at risk at baseline. When the outcomes are COVID-19-related hospitalization, LOS, and death, the study sample should be those who tested positive; when the outcome is ICU admission, the sample should be those who were hospitalized due to COVID-19 (assuming that ICU admission were all transferred from regular admissions).

ANSWER: Thank you for the opportunity to discuss this topic. The denominator of the analysis of COVID-19 related hospitalization, ICU admission and deaths is the whole cohort (n=4,660,502), while for hospital length-of-stay, the denominator was everyone hospitalized. We discussed this issue during our protocol and analysis plan. We discuss the reasons to analyse the whole cohort below:

1 – To estimate the association in the target population in which all individuals are at risk.¹² Our outcome is a composite/conditional outcome: being diagnosed with COVID-19 plus having a severe event.¹² Those that have not been diagnosed are still at risk of the composite outcome, because they are at risk of being infected, then at risk of having the severe event within 30 days (COVID-19 diagnosis plus severe event). As our goal was to derive estimates for the target population, which was the full population of Catalonia. Similarly, we aimed to estimate mortality rates (for analysis focused on deaths) rather than case-fatality rates, to increase the broader public health relevance of the findings.¹²

2- To avoid an expected and likely important selection bias when analysing only COVID-19 cases: Particularly for the first wave, and even in the second wave, the number of those undiagnosed with COVID-19 is high.¹³ Probability of testing is likely associated with air pollution levels (in our cohort, those diagnosed with COVID-19 have higher levels of air pollution) and there are likely unmeasured factors associated with both, this gives risk to selection/collider bias. These issues have been discussed in these two methodological papers,^{14,15} which recommend against restricting analysis to those diagnosed with COVID-19. As we don't expect severe COVID to be undiagnosed (people will go to hospital if they are very ill), we select to analyse severe COVID-19.

3 – A large proportion of hospitalized individuals had their diagnosis upon hospitalization: 14,556 (31%) of the 47,174 hospitalizations had their first COVID-19 diagnosis at the time of hospitalization. These individuals would have 0 time to event if we included only diagnosed individuals. These individuals could be those at higher risk, or have less access to primary care, which is likely associated with the exposure, potentially creating an additional source of selection bias.

We evaluated the results for hospitalization, ICU admission and deaths among those with a COVID-19 diagnosis. We present the results for all COVID-19 diagnosis, attributing 0.5 to time to event for those with event same day of diagnosis; and an analysis among those diagnosed through primary care (**supplementary tables S9, S10, S11**). As can be observed, there are still positive associations for hospital and ICU admissions, and depending on the model, for death. In wave-specific analyses, there is an impact of wave particularly for death, reflecting our previous arguments that the analysis of COVID-19 cases can be biased, particularly during the first wave.

The estimates for O₃ are confusing and may be misleading. It is hard to interpret O₃ as some sort of "good air". What are the correlation coefficients between PM_{2.5}, NO₂, and O₃? I'm not sure if it is a good idea to put the results of O₃ in a paper that is primarily targeting PM_{2.5} and NO₂.

ANSWER: Thanks for this comment. Following the editor and reviewers' comments, we present all O₃ results in the supplement. We agreed it is difficult to interpret the O₃ coefficients given the high negative correlation with the other pollutants. We highlight the issue of potential confounding by correlated co-pollutants in the revised version of the manuscript, particularly in the interpretation of changes in the O₃ estimates when co-adjusting by NO₂.

The correlation between the air pollutants is in the supplementary appendix and shown below.

PM_{2.5} x NO₂: 0.89

PM_{2.5} x O₃: -0.76

NO₂ x O₃: -0.82

Minor comments:

Page 6, lines 125-128, the definition of a clinical diagnosis of COVID-19 is not clear to me. Could the authors provide more details about this?

ANSWER: Clinical diagnosis was based on the ICD-10 codes in the health system, which may or may not have included testing for SARS-COV2 infection. We clarified this point.

Page 8, lines 163-166, what is the difference between previous chronic comorbidities and individual health risk groups? They were both based on the individual's chronic

comorbidities. Claiming acute myocardial infarction as a chronic comorbidity does not make sense to me (acute is an antonym to chronic).

ANSWER: Thanks for this comment. Individual health risk groups are a composite index with more than 31 items and chronic comorbidities which are weighted to derive the score. The score is focused on the multi-comorbidity (not only the chronic comorbidity per se), and includes acute comorbidities, pregnancy, health system demand and health access. We added a sentence in the discussion about this issue. More information is in the supplementary methods.

We used acute myocardial infarction codes to capture those who had an AMI and survived as a chronic comorbidity indicator of living with advanced atherosclerosis or coronary disease. We agree using this example was not the best one and changed it to chronic obstructive pulmonary disease (COPD).

Is it possible to tell different COVID variants for the 340,608 COVID-19-positive individuals in the analysis?

ANSWER: We don't have individual sequenced data. Since they are all cases from 2020, therefore they are very likely the original strain and part the 20E (EU1) variant of interest that appeared during the 2020 summer in Spain.¹⁶

Page 15, line 321: please refrain from using "RR" for logistic regression or generalized linear models, they are not the same thing (even though they are close in certain situations).

ANSWER: Thanks for this, we corrected it to OR.

Page 19 408-409: what is the general composition of race/ethnicity? and what are the potential impacts on the findings of this study?

ANSWER: We don't have reliable data about the race/ethnicity distribution in Catalonia. In our cohort, we don't have this data linked at the individual level. Looking at the Catalonian registry, defined by nationality, in 2015, the non-Spanish population in Catalonia was approximately 1.0 million individuals (~290,000 from Africa, ~280,000 from other countries in the European Union, ~180,000 from South America and ~140,000 from Asia and Oceania, etc).

Reviewer #3 (Remarks to the Author):

The paper is very well-written. Some key strengths of this study include its use of state-of-the-art methods for air pollution exposure assessment and the population-based cohort study design that consisted of nearly entire adult population in Catalonia, Spain. The findings suggest that long-term exposures to PM_{2.5} and NO₂ were associated with severe COVID-19 outcomes in this cohort. The evidence concerning O₃ exposure is less clear. The authors are commended for conducting a number of sensitivity analyses that resulted in similar findings indicating the robustness of the results. Although several individual-level factors such as race/ethnicity and occupation could not be controlled for, the set of other covariates was ample.

I only have few minor comments and suggestions that are listed below.

ANSWER: Thank you for your comments.

Methods/Covariates

1. For many readers who are unfamiliar with Catalonia, please provide more details about this study area, especially the size of various geographic regions used in this study (eg, health regions and primary care service areas). It would helpful to also discuss the appropriateness of adjusting for variables derived at these geographic levels.

ANSWER: We added more information about these divisions in the methods section and maps in supplementary material. The specific area that was used for these variables was largely driven by data availability. In general, we used the smallest available area unit.

2. Could the minimal impact of further adjusting for preexisting comorbidities on the air pollution-COVID-19 associations be due to the fact that the main model already included health risk group? As it was described, the latter variable encompassed multimorbidity. Please clarify.

ANSWER: We agree with the reviewer that the health risk group could, at least in part, account for the preexisting comorbidities adjustment. We highlighted this in the discussion section of the revised version.

3. As the primary outcome of interest is COVID-19 hospitalizations, could the authors consider a sensitivity analysis to adjust for distance to a nearest acute-care hospital?

ANSWER: We run a sensitivity analysis accounting to the distance to the nearest acute-care hospital (**supplementary tables S3, S4, S5**). The results were robust to this adjustment.

4. Page 9. The authors “accounted for the competing event of death when evaluating COVID-19 related hospitalization and ICU admission by censoring at a death event”. Since ambient air pollution has been linked to various mortality outcomes, the censoring may introduce selection bias under the null. Could the authors consider a sensitivity analysis to account for the censoring using the measured risk factors for the outcome of interest (for example, by IP censoring weighting)?

ANSWER: Thanks for the suggestion. We accounted for death in the framework of competing risk events using causal-specific Hazard Ratios (which is applied as censoring as in a Cox PH model). To use inverse probability censoring weighting is similar to what Fine & Gray described in their model to estimate subdistribution Hazard Ratios and cumulative incidence functions (CIF). Therefore, the impact of death is already accounted for in our modeling, including the effect of measured risk factors in the model.¹⁷⁻¹⁹ Assuming we have a null cause-HR on COVID-19 hospitalizations and a positive cause-HR on deaths, using a subdistribution approach could result in a subdistribution-HR < 1 for COVID-19.¹⁷ At the same time, any potential diversion from the current results is expected to be minimal, because from the 4,660,502 individuals followed during 2020, there were 49,331 deaths. From these, only 41,303 deaths (3,744 COVID-19 deaths and 37,559 non COVID deaths) occurred before the event of interest (COVID-19 hospitalization), being censored during the follow-up.

Discussion

5. Page 18. The authors discussed “Another hypothesis is that air pollution exposure could facilitate SARS-CoV2 binding, because there is evidence that exposure to particulate matter upregulates the expression of SARS-CoV-2 receptors in the lung (e.g., angiotensin-converting enzyme 2)”. This manuscript can benefit from a more thorough discussion about this possible mechanism, since its results suggested that the impact of air pollution on increasing the prevalence of chronic comorbidities associated with severe COVID-19 was likely minimal. This will help improve the biological plausibility of the study results.

ANSWER: We agree with the reviewer and reinforced this idea in the discussion. We refer to this as one of the postulated mechanisms but it was not something we could test directly in our analysis.

References:

1. Reis, G. *et al.* Early Treatment with Pegylated Interferon Lambda for Covid-19. *N Engl J Med* **388**, 518–528 (2023).
2. Avezum, Á. *et al.* Hydroxychloroquine versus placebo in the treatment of non-hospitalised patients with COVID-19 (COPE – Coalition V): A double-blind, multicentre, randomised, controlled trial. *The Lancet Regional Health - Americas* **11**, 100243 (2022).
3. Butler, C. C. *et al.* Molnupiravir plus usual care versus usual care alone as early treatment for adults with COVID-19 at increased risk of adverse outcomes (PANORAMIC): an open-label, platform-adaptive randomised controlled trial. *The Lancet* **401**, 281–293 (2023).

4. More information on data sources related to coronavirus (COVID-19) - Office for National Statistics.
<https://www.ons.gov.uk/peoplepopulationandcommunity/healthandsocialcare/conditionsanddiseases/articles/moreinformationondatasourcesrelatedtocoronaviruscovid19/2020-12-11#deaths>.
5. Nobile, F. *et al.* Air pollution, SARS-CoV-2 incidence and COVID-19 mortality in Rome – a longitudinal study. *Eur Respir J* 2200589 (2022) doi:10.1183/13993003.00589-2022.
6. Chen, Z. *et al.* Ambient Air Pollutant Exposures and COVID-19 Severity and Mortality in a Cohort of COVID-19 Patients in Southern California. *Am J Respir Crit Care Med* rccm.202108-1909OC (2022) doi:10.1164/rccm.202108-1909OC.
7. Cajanding, R. J. M. Silent Hypoxia in COVID-19 Pneumonia: State of Knowledge, Pathophysiology, Mechanisms, and Management. *AACN Advanced Critical Care* **33**, 143–153 (2022).
8. Groenwold, R. H. H. *et al.* Missing covariate data in clinical research: when and when not to use the missing-indicator method for analysis. *CMAJ* **184**, 1265–1269 (2012).
9. Shaaban, A. N., Peleteiro, B. & Martins, M. R. O. Statistical models for analyzing count data: predictors of length of stay among HIV patients in Portugal using a multilevel model. *BMC Health Serv Res* **21**, 372 (2021).
10. Fernandez, G. A. & Vatcheva, K. P. A comparison of statistical methods for modeling count data with an application to hospital length of stay. *BMC Med Res Methodol* **22**, 211 (2022).
11. Austin, P. C., Rothwell, D. M. & Tu, J. V. A Comparison of Statistical Modeling Strategies for Analyzing Length of Stay after CABG Surgery. *Health Services and Outcomes Research Methodology* **3**, 107–133 (2002).
12. Westreich, D., Edwards, J. K., Tennant, P. W. G., Murray, E. J. & van Smeden, M. Choice of Outcome in COVID-19 Studies and Implications for Policy: Mortality and Fatality. *American Journal of Epidemiology* **191**, 282–286 (2022).

13. Pollán, M. *et al.* Prevalence of SARS-CoV-2 in Spain (ENE-COVID): a nationwide, population-based seroepidemiological study. *The Lancet* **396**, 535–544 (2020).
14. Griffith, G. J. *et al.* Collider bias undermines our understanding of COVID-19 disease risk and severity. *Nat Commun* **11**, 5749 (2020).
15. Millard, L. A. C. *et al.* Exploring the impact of selection bias in observational studies of COVID-19: a simulation study. *International Journal of Epidemiology* **52**, 44–57 (2023).
16. Hodcroft, E. B. *et al.* Spread of a SARS-CoV-2 variant through Europe in the summer of 2020. *Nature* **595**, 707–712 (2021).
17. Lau, B., Cole, S. R. & Gange, S. J. Competing Risk Regression Models for Epidemiologic Data. *American Journal of Epidemiology* **170**, 244–256 (2009).
18. Putter, H., Schumacher, M. & van Houwelingen, H. C. On the relation between the cause-specific hazard and the subdistribution rate for competing risks data: The Fine-Gray model revisited. *Biom. J.* **62**, 790–807 (2020).
19. Austin, P. C. & Fine, J. P. Practical recommendations for reporting Fine-Gray model analyses for competing risk data. *Statistics in Medicine* **36**, 4391–4400 (2017).

REVIEWERS' COMMENTS

Reviewer #1 (Remarks to the Author):

All of my comments have been addressed. Thank you

Reviewer #2 (Remarks to the Author):

The authors have fully addressed my comments.